# Variability of amylose content and its correlation with the paste properties of cassava starch

**Natalia Rocha Ribeiro[1], Massaine Bandeira e Sousa[2], Luciana Alves de Oliveira[2], Eder Jorge de Oliveira**[2]*

**1** Universidade Federal do Recôncavo da Bahia, Cruz das Almas, Bahia, Brazil, **2** Embrapa Mandioca e Fruticultura, Cruz das Almas, Bahia, Brazil

* eder.oliveira@embrapa.br

**Data Availability Statement:** All relevant data are within the manuscript and its Supporting information files.

**Funding:** Natalia Rocha Ribeiro: CAPES (Coordenação de Aperfeiçoamento de Pessoal de

## Abstract

The amylose content can significantly impact the diverse industrial applications of cassava starch. This study aimed to assess the variability of cassava germplasm concerning amylose content and other attributes pertinent to root quality, alongside its correlation with paste properties. Starch extracted from 281 genotypes, obtained in germplasm evaluation trials, was evaluated for amylose content, with additional analysis of parameters such as pasting temperature, time to peak viscosity (TPV), viscosity breakdown (BrD), retrogradation tendency, and maximum, minimum, and final viscosities. The genotypes exhibited considerable variation in dry matter content (ranging from 27.06% to 41.02%), starch content (from 14.61% to 25.67%), cyanogenic compounds (1.77 to 7.81), and amylose content (0.05% to 33.23%). High phenotypic variability in paste properties was observed, alongside a low residual effect for most traits, resulting in high broad-sense heritabilities (>0.95). Strong correlations of significant magnitude (>0.80) were found between parameters such as peak viscosity × viscosity breakdown, minimum viscosity × final viscosity, and final viscosity × retrogradation tendency. Moderate correlations were also identified, such as between dry matter content × starch content (0.56). While positive, correlations between amylose content and paste properties were of low magnitude (ranging from 0.13 to 0.35), except for TPV and BrD. Principal component discriminant analysis clustered the germplasm into six distinct groups based on root quality and paste properties, with most improved genotypes falling into two clusters characterized by high starch and dry matter contents. This study underscores the necessity of simultaneous evaluation of amylose content and paste properties in the breeding pipeline. Additionally, clustering cassava genotypes proves beneficial in identifying those that fulfill specific requirements in industrial and breeding applications.

## 1. Introduction

Cassava (*Manihot esculenta* Crantz) is a staple crop, and therefore, a food security specie for millions of people in tropical and subtropical regions worldwide, ranking as the fourth most

Nível Superior). Grant number: 88887.641163/2021-00; Massaine Bandeira e Sousa: Empresa Brasileira de Pesquisa Agropecuária. Grant number: 20.18.01.012.00.00; Eder Jorge de Oliveira: CNPq (Conselho Nacional de Desenvolvimento Científico e Tecnológico). Grant number: 409229/2018-0, 442050/2019-4 and 303912/2018-9; Eder Jorge de Oliveira: FAPESB (Fundação de Amparo à Pesquisa do Estado da Bahia). Grant number: Pronem 15/2014. This work was partially funded by UK's Foreign, Commonwealth & Development Office (FCDO) and the Bill & Melinda Gates Foundation. Grant INV-007637. Under the grant conditions of the Foundation, a Creative Commons Attribution 4.0 Generic License has already been assigned to the Author Accepted Manuscript version that might arise from this submission. The funder provided support in the form of fellowship and funds for the research, but did not have any additional role in the study design, data collection and analysis, decision to publish, nor preparation of the manuscript.

**Competing interests:** The authors have declared that no competing interests exist.

important staple food after rice, wheat, and maize [1, 2]. The species holds significant economic importance as one of the primary sources of starch, alongside maize, potato, and rice [3, 4], displaying reduced syneresis and a neutral flavor, thus providing a competitive edge in the food industry over cereal starches [5].

Industrial applications of starch rely on paste properties, including gelatinization [6–8], which transforms starch into a viscous paste with increasing temperature. Understanding the optimal water quantity and temperature is crucial, varying according to the type of starch [9].

The viscosity of starch suspensions is usually assessed using viscoamylographs. Peak viscosity of cassava starch (ranging between 146–1505 cP), breakdown (ranging between 28–859 cP), and retrogradation tendency (ranging between -702 to 273 cP) are critical points, influencing the quality and performance of starch in various industrial and culinary applications [10].

Santos et al. [11] reported that cassava starch exhibits a lower gelatinization temperature and reduced retrogradation tendency compared to cereal starches. This characteristic implies greater stability and a diminished likelihood of clumping or stickiness when used as a thickener or stabilizer in food products. Additionally, cassava starch is noted for its resistance to breakdown during industrial processing, resulting in final products with improved texture and appearance [10]. Carmo et al. [5] and Sakurai et al. [12] further highlight various advantages of cassava starch, including its suitability as a gluten-free alternative for individuals with celiac disease and its high clarity when forming pastes. Moreover, the high purity of cassava starch makes it a preferred choice for use in the food industry.

Within the same species, there exist variations in paste properties and amylose content, offering opportunities for the selection of new varieties to be explored for various industrial purposes [13]. Generally, some researchers suggest that starches with high amylose contents are better suited in the manufacture of biofilms [21], production of adhesives and as binders [34] while those with low amylose contents are more commonly employed in the frozen food [11]. Therefore, alongside numerous chemical modifications targeting specific properties, genetic variability can be leveraged in the development of cultivars with naturally modified starches.

In recent years, numerous studies have documented variability in paste properties among cassava starches [10, 14–19]. Sánchez et al. [10] provided a comprehensive assessment by evaluating approximately 4,000 cassava genotypes from the International Center for Tropical Agriculture (CIAT), revealing significant variability in peak viscosity (146–1,505 cP), breakdown (28.1–859 cP), and retrogradation tendency (-702–273 cP) within the CIAT germplasm. In another recent study in Brazil, Santos et al. [17] assessed over 1000 cassava germplasm accessions and reported distinct variations in peak viscosity (3,279–6,459 cP), breakdown (134–848 cP), and retrogradation tendency (290–3,681 cP). Generally, starches with low retrogradation tendencies are preferred as gelling agents in refrigerated and frozen foods, whereas those with high swelling power (temperature ranging from 62.6 to 70.1 °C) are recommended for use as thickeners and emulsifiers in food products [16]. Thus, the considerable variability present in cassava germplasm facilitates the selection and development of varieties with added value for various agri-food purposes.

The ability of starch granules to form a paste post-gelatinization is primarily attributed to the presence of amylose [20, 21], which plays a pivotal role in determining the physicochemical properties of starch, including its paste characteristics [6, 8]. Normal or native cassava starches typically contain an average of 21% amylose, with a range spanning from 13% to 29% [10, 20]. While waxy cassava mutants with starch [low amylose] or and smaller starch granules have been identified [22], there are no documented instances of cassava with notably high amylose content (>40%).

Significant correlations between paste properties and amylose content have been established in various crops, including red rice [23], maize [24], potato [25], wheat [26], and sorghum [27]. Karakelle et al. [24] noted that starches with high amylose content exhibit substantially lower viscosity values and higher pasting temperatures compared to those with low amylose content. Conversely, in sorghum, amylose content displayed negative correlations with peak viscosity or maximum viscosity and positive correlations with retrogradation tendency and cold paste viscosity [27]. These findings suggest that optimizing the amylose/amylopectin ratio can be achieved without the need for modifying starch properties to attain products with the desired characteristics.

Despite the crucial role of amylose in starch functionality, the correlation between paste properties and amylose content in cassava remains underexplored but is vital for guiding genetic improvement efforts. Therefore, comprehensive studies involving diverse populations exhibiting significant amylose content variations are imperative for accurately delineating its relationship with paste properties, with the goal of developing more promising industrial starch varieties [6]. Thus, this study aims to analyze the genetic diversity within the cassava germplasm bank concerning amylose content, investigating its association with starch paste properties in a population showcasing substantial variations in root yield and attributes.

## 2. Materials and methods

### 2.1 Evaluation of cassava root quality-associated traits

A total of 30 trials evaluating clones from the Embrapa breeding program were conducted in the cities of Cruz das Almas, Laje, and Valença in Bahia, Brazil, from 2011 to 2022. Detailed information regarding the soil type at each evaluation site is presented in Table 1. According to the Köppen classification, the climate in the region is type Af, indicating a hot and humid tropical climate, with an average annual precipitation ranging from 1,200 to 1,500 mm, highest rainfall occurring from March to July, average annual temperature of 24.5 ºC, and average relative humidity of approximately 80%.

In total, 281 genotypes were evaluated in the field (S1 Table) using an augmented block design, with plots consisting of two rows with eight to ten plants each (totaling 16 to 20 plants per plot), spaced 0.90 m between rows and 0.80 m between plants. Between 4 and 34 improved cultivars from Embrapa and local cultivars were planted as common checks in different augmented blocks (ranging from 5 to 22). Conventional farming practices were employed in the region, involving soil preparation (one plowing, two harrowings, and opening of planting furrows). The cassava cuttings material was planted with approximately 16 to 18 cm in length.

All field trials were conducted under rainfed conditions (without supplementary irrigation), following local crop management practices, in accordance with the recommendations of Souza et al. [28]. Planting was carried out during the rainy season of the region (May to

**Table 1. Information about the locations where trials for evaluating root quality traits were conducted, from 2011 to 2022.**

| Trial | City / state | Altitude | Soil type | N# trials |
|---|---|---|---|---|
| CNPMF | Cruz das Almas, BA | 215 | Yellow Latosol | 14 |
| CNPMF Area 2 | Cruz das Almas, BA | 215 | Yellow Latosol | 1 |
| Coopamido | Laje, BA | 180 | Red Yellow Latosol | 4 |
| Novo Horizonte | Laje, BA | 180 | Red Yellow Latosol | 2 |
| Novo Rumo | Laje, BA | 180 | Red Yellow Latosol | 1 |
| UFRB-PP1 | Cruz das Almas, BA | 215 | Yellow Latosol | 5 |
| UFRB-Candial | Cruz das Almas, BA | 215 | Yellow Latosol | 3 |

August) to ensure the minimum soil moisture required for germination and crop establishment.

Harvesting of the trials was done manually between 10 to 12 months after planting. The following root quality traits were evaluated: 1) dry matter content (DMC, %) in roots based on the gravimetric method: approximately 5 kg of roots were cleaned to remove excess soil, and their tips were trimmed. Then, the weight of roots in air and in water was measured, and DMC was calculated using the formula: $DMC = 158,3 \times \frac{weight\ in\ air}{weight\ in\ air - weight\ in\ water} - 142$, as described by Kawano et al. [29]; 2) starch content (SC, %) in roots: Starch from 500 g of roots from different plants in the plot was extracted as described in section 2.2.; cyanogenic compounds content (HCN): determined using the picrate method [30].

## 2.2 Starch extraction

The set of cassava samples analyzed consisted mainly of genotypes with significant variation in cyanogenic compound content (HCN), as well as in relation to dry matter content (DMC) and starch content. For this analysis, a panel containing 281 cassava clones, including improved and local (unimproved) cultivars (S1 Table), was selected. Most of these samples originated from Brazil.

Starch extraction followed the procedure outlined by Vasconcelos et al. [31]. Cassava roots were chosen, washed, and peeled to eliminate any external debris or impurities. Next, the root samples were sliced into smaller fragments to ease processing. These fragments were then blended at a ratio of 500 g of root to 500 ml of cold water (~5˚C), with processing alternating between 1 min of blending, 1 min of rest, and an additional 1 min and 30 s of blending. The resulting blend underwent filtration using voile fabric and a sieve with a mesh size of 220 micrometers. Subsequently, 500 ml of cold water (5 ˚C -in unit) was added to the filtered mass, which was blended again for approximately 1 min. The filtered mass was rinsed with 3 liters of chilled water. The liquid component was left to settle in a cold chamber (around ±5 ˚C) for 12 h to facilitate starch sedimentation, separating the liquid from the solid starch. After this period, the supernatant was discarded, and the starch was lightly washed with 20 ml of 95% alcohol. Following this, the starch was dried in an oven (NE, Brazil, Model 420–7) at approximately ±40 ˚C for 3 days. Finally, the dried starch underwent a maceration process using a mortar and pestle until it reached a fine powder consistency (< 0,1 mm). The starch powder was then stored in airtight plastic containers, adequately shielded from moisture and potential sources of contamination, at approximately ±5 ˚C.

## 2.3 Evaluation of starch paste properties

The starch paste properties were assessed using a rapid viscosity analyzer (model RVA-4500, series 4; Newport Scientific, Warriewood, Australia), employing Standard 1 programming on Thermocline for Windows (TCW) software, version 7 (NewPort Scientific, Australia).

The analysis was conducted in duplicate using 3 g of starch sample (±14% moisture) in 25 g of distilled water. Correction of the samples and water weights to achieve 14% moisture content was performed using an infrared moisture analyzer (Gehaka, Brazil, model IV2500). For this, the following formulas were applied: $m2 = \frac{(100-14) \times m1}{(100-u1)}$ and $w = 25,0 + (m1 - m2)$, where $m1$ is the mass of the sample in grams; $m2$ is the mass of the sample corrected to 14% basis in grams; $u1$ is the moisture content of the sample in %; and $w$ is the corrected water weight in grams.

The starch sample was added to the container with water and homogenized with the RVA helix to prevent lump formation. The starch suspension underwent the following temperature/

time program: heating to 50 ˚C for 1 min; heating from 50 to 95 ˚C at a rate of 6 ˚C per min; holding at 95 ˚C for 2.5 min; cooling from 95 ˚C to 50 ˚C at a rate of 6 ˚C per min. Finally, the sample was held at 50 ˚C for 2 min. The suspension was subjected to 960 rpm for the first 10 s of the test and 160 rpm for the remainder of the time. The total analysis duration was 13 min per sample replicate. The evaluated parameters included: pasting temperature in ˚C (PaT); peak viscosity in cP (PeV); minimum viscosity at 95 ˚C in cP (MiV); viscosity breakdown in cP (BrD) which is obtained by the formula BrD = minimum viscosity—peak viscosity; final viscosity in cP (FiV); and setback or retrogradation tendency in cP (SeB), which is obtained by the formula SeB = final viscosity—minimum viscosity.

## 2.4 Evaluation of amylose content

To determine the amylose/amylopectin ratio, the iodine colorimetric method [32] was employed, which is based on the light transmission through the complex formed by the reaction between amylose and iodine. Starch granules were dispersed with ethanol and gelatinized with 1 N sodium hydroxide. An aliquot was acidified, and after reaction with iodine, the blue-colored complex was quantified by spectrophotometry at 620 nm (Biospectro, model SP 220, Brazil). A standard curve was constructed using diluted and serially diluted solutions of standard amylose (CAS: 9005-62-7, Sigma-Aldrich, St. Louis, USA) and amylopectin (waxy starch extracted from cassava root with 0% amylose). Results were expressed as a percentage according to the formula: Amylose $(\%) = \frac{y \times 0.1}{m} \times 100$, where $y$ represents the mass of amylose, 0.1 is the mass of the sample in grams that should be weighed to obtain the same result as the calibration curve, and $m$ is the mass of the sample in grams.

## 2.5 Data analysis

The amylose content and starch paste properties were subjected to analysis of variance (ANOVA), using the following linear model: $Y_{ik} = \mu + g_i + \epsilon_{ik}$, where $Y_{ik}$ represents the phenotypic observation of genotype $i$ in repetition $k$; μ is the overall mean; $g_i$ is the fixed effect of genotype, and $\epsilon_{ik}$ is the random residual effect of genotype $i$ in repetition $k$. The data related to root quality were subjected to mixed model analysis for estimating the Best Linear Unbiased Estimators (BLUEs), using the following model: $Y_{ijk} = \mu + g_i + l_j + a_k + (g \times l)_{ij} + (g \times a)_{ik} + (g \times l \times a)_{ijk} + \epsilon_{ij}$, where $Y_{ijk}$ is the phenotypic observation of genotype $i$ within location $j$ and year $k$; μ is the overall mean, $l_j$ is the random effect of location; $a_k$ is the random effect of year; $g_i$ is the fixed effect of genotype, $(g \times l)$ is the random effect of genotype-location interaction, $(g \times a)$ is the random effect of genotype-year interaction, $(g \times l \times a)$ is the random effect of genotype-location-year interaction, and $\epsilon_{ijk}$ is the residual random effect of genotype $i$ within location $j$ and year $k$.

The Pearson correlation between amylose content and starch paste properties was analyzed using the corrplot package [33]. Additionally, Principal Component Discriminant Analysis (DAPC) was performed using the adegenet package [34]. The number of clusters based on root quality attributes and agronomic aspects was determined using successive K-means clustering with an increasing number of groups (k ranging from 2 to 15) after transforming the dataset by Principal Component Analysis using the find.clusters function of the adegenet package [34]. The Bayesian Information Criterion (BIC) was used to define the ideal number of clusters and the most suitable clustering solution with the lowest BIC. After determining the most suitable number of groups to represent germplasm diversity, the dendrogram was constructed using the circlize package [35]. Boxplots were obtained using the ggstatsplot package [36] to visualize the distribution and differences between clusters, compare different groups or variables, identify patterns, and provide a quick overview of descriptive statistics of the dataset

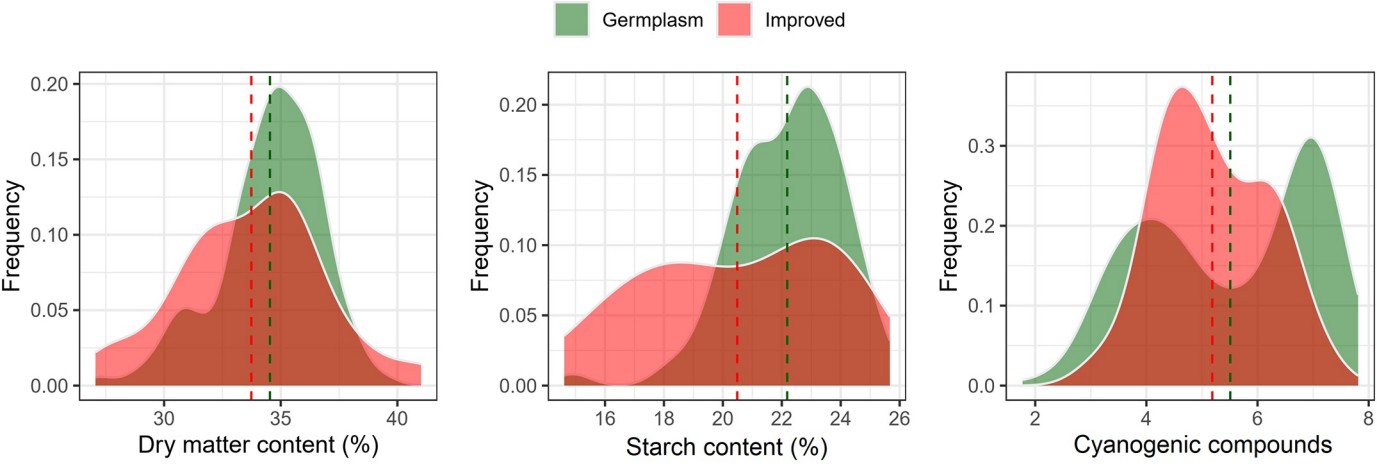

**Fig 1. Distribution of best linear unbiased estimator (BLUEs) of dry matter content (DMC, %), starch (StC, %), and cyanogenic compounds (HCN) in cassava genotypes.** Red color: genotypes belonging to the improved population; Green color: genotypes belonging to the germplasm bank.

of the 281 cassava genotypes under analysis. All packages were loaded and analyzed using R software, version 4.2.3 [37].

The selection index based on the sum of ranks [38] and predefined weights was used to select genotypes with a better balance of amylose content and root quality, according to the formula: SI = (DMC × 10)+(Tamido × 10)+(HCN ×−30) which refers to the sum of the BLUEs of each characteristic multiplied by their respective economic weights. Initially, genotypes were pre-selected based on their amylose content values, i.e., low (<12%) and high content (>25%). Additionally, the selection differential was calculated based on the difference between the overall mean for each group and the mean of the selected genotypes.

## 3. Results

### 3.1 Variation of genetic parameters of root quality, amylose, and paste properties

The distribution of BLUEs for root quality traits by population (genetic material origin) is presented in Fig 1. The range of variation of BLUEs for dry matter content was quite wide (27.06 to 41.02%), with an average of 33.74% (improved genotypes) and 34.54% (germplasm). The same trend was observed for starch content with a variation of 14.61 to 25.67% (average of 20.49% in improved genotypes and 22.18% in germplasm). For HCN, the range of variation was also quite high, ranging from score 1.77 to 7.81, with similar averages between the two populations (score 5.19 and 5.51). Only for HCN, there was a bimodal distribution of genotypes, especially those belonging to germplasm, and a frequency of genotypes with higher HCN content. For the other traits, there was a distribution close to normality (Fig 1).

Similarly to the previous traits, there was a wide phenotypic variation for amylose content and starch paste properties (Fig 2). The variation in amylose content ranged from 0.05 to 33.23%, with an average of 19.8% in improved genotypes and 22.1% in germplasm. Based on the genotypes evaluated in this study, no cassava genotypes with amylose content ranging between 5 and 10% were identified.

All parameters associated with paste properties showed high phenotypic variation among the evaluated genotypes, such as peak or maximum viscosity (ranging from 2836.5 to 6676.5 cP with a mean of 4413 cP in improved genotypes and 4313 cP in germplasm), minimum

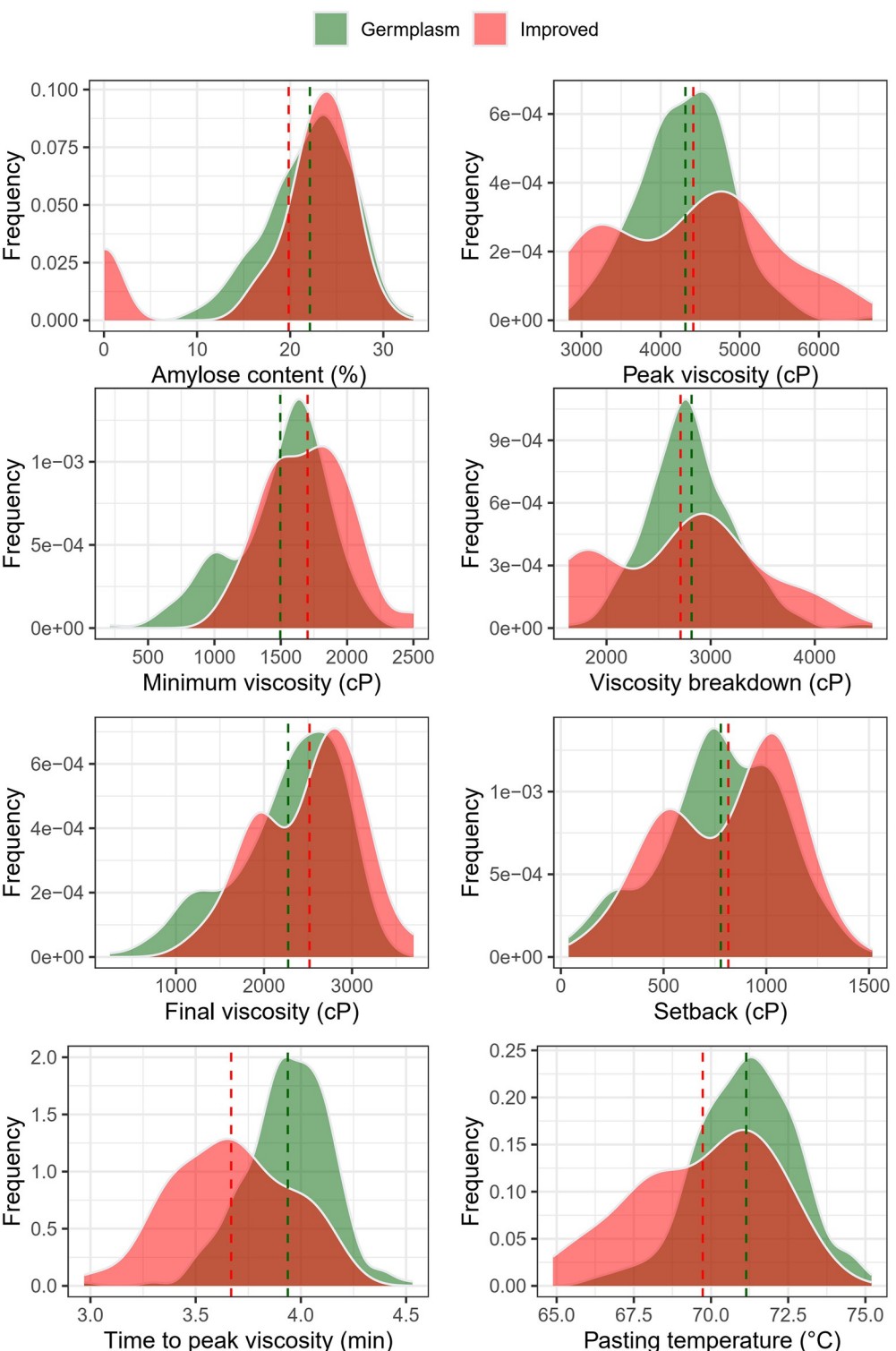

**Fig 2. Frequency distribution of amylose content and various paste properties among cassava genotypes.**
Genotypes from the improved population are represented in red, while those from the germplasm bank are in green.

**Table 2. Analysis of variance, phenotypic mean, and broad-sense heritability of amylose content and paste properties of cassava starch samples.**

| Source of variation | DF | Mean square | | | | | | | |
|---|---|---|---|---|---|---|---|---|---|
| | | **PeV** | **MiV** | **BrD** | **FiV** | **SeB** | **TPV** | **PaT** | **Aml** |
| Genotype | 280 | 826170.8* | 257239.2* | 440669.6* | 764476.5* | 177495.5* | 0.1* | 6.2* | 81.3* |
| Rep | 1 | 3872.0 | 107.3 | 2690.3 | 76.3 | 364.5 | 0.0 | 0.0 | 1.2 |
| Residue | 296 | 23928.0 | 3760.0 | 17425.0 | 9011.0 | 2975.0 | 0.0 | 0.2 | 1.8 |
| $H^2$ | | 0.97 | 0.99 | 0.96 | 0.99 | 0.98 | 0.95 | 0.96 | 0.98 |
| Mean | | 4325.6 | 1525.0 | 2800.6 | 2310.9 | 785.8 | 3.9 | 71.0 | 21.9 |
| CV% | | 0.0 | 0.0 | 0.0 | 0.0 | 0.1 | 0.0 | 0.0 | 0.1 |

DF: degree of fredoom; PeV: Peak viscosity (cP); MiV: Minimum viscosity (cP); BrD: Viscosity breakdown (cP); FiV: Final viscosity (cP); SeB: Retrogradation tendency (cP); TPV: Time to peak viscosity (min); PaT: pasting temperature (˚C); Aml: Amylose content (%)

viscosity (ranging from 506.0 to 2501.5 cP with a mean of 1702 cP in improved genotypes and 1496 cP in germplasm), viscosity breakdown (ranging from 1636.5 to 4552.0 cP with a mean of 2711 cP in improved genotypes and 2817 cP in germplasm), final viscosity (ranging from 251.0 to 3699.5 cP with a mean of 2517 cP in improved genotypes and 2275 cP in germplasm), retrogradation tendency (ranging from 38 to 1516 cP with a mean of 815 cP in improved genotypes and 779 cP in germplasm), time to peak viscosity (ranging from 2.97 to 4.53 min with a mean of 3.67 min in improved genotypes and 3.94 min in germplasm), and pasting temperature (ranging from 64.88 to 75.2 ˚C with a mean of 69.7 ˚C in improved genotypes and 71.1 ˚C in germplasm) (Fig 2). The improved population exhibited, on average, lower amylose values.

There was a significant effect ($p < 0.01$) for all traits related to amylose content and paste properties (Table 2). Since the residual effect and coefficient of variation estimates were very low, the heritability estimates were quite high for all these characteristics ($H^2 \geq 0.95$).

## 3.2 Phenotypic correlations

Positive phenotypic correlations between root quality traits, amylose content, and paste properties vary between 0.12 and 0.95, while negative correlations range from -0.14 to -0.41 (Fig 3). The highest positive correlations (>0.80) were observed between peak viscosity and viscosity breakdown, minimum viscosity and final viscosity, and final viscosity and retrogradation tendency. Moderate positive correlations (0.50–0.80) are evident for dry matter content versus starch content, peak viscosity versus minimum viscosity and final viscosity, and peak viscosity and minimum viscosity versus retrogradation tendency. Correlations between amylose content and paste properties are positive but of low magnitude (0.14–0.35), except for the correlation between amylose content and minimum viscosity, which is not statistically significant.

Genotypes were categorized into two populations based on their status within the breeding program: those belonging to the germplasm bank (BAG) and those undergoing or having undergone improvement (improved population). Significant positive phenotypic correlations were observed within the germplasm for dry matter content and HCN, while a negative correlation was observed between starch content and amylose content (-0.18) (Fig 3). Correlations between these traits in improved genotypes did not reach significance, possibly due to the smaller sample size compared to the germplasm. In the improved population, correlations between amylose content and paste properties (PeV, MiV, BrD, FiV, SeB, and TPV) were stronger (0.60–0.86) compared to those observed in the germplasm.

Correlations between starch quality traits and amylose content were assessed considering genotypes' classification according to their HCN content (Fig 4). Genotypes classified as sweet

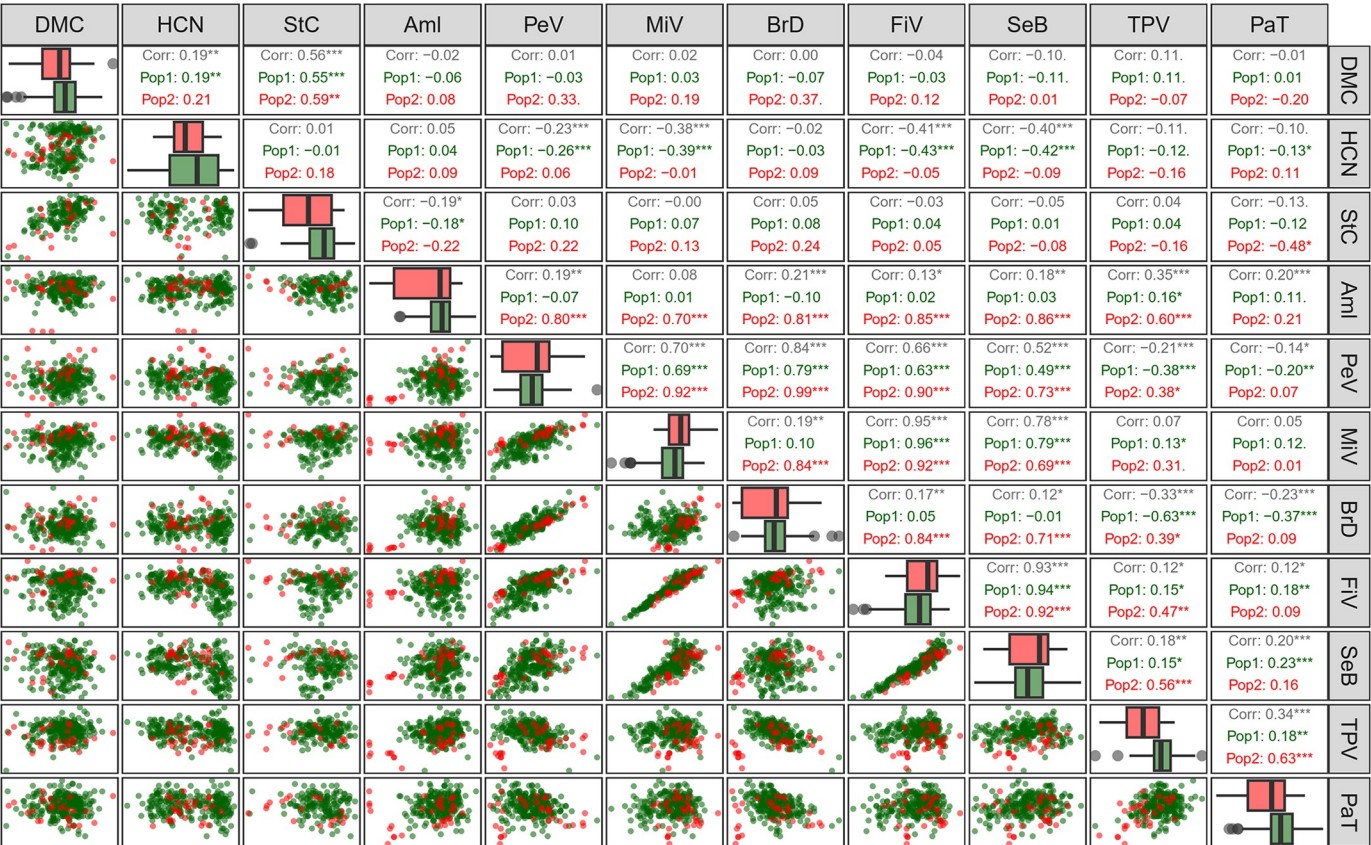

**Fig 3. Pearson correlation by population (Pop1: germplasm; Pop2: improved population) among root quality traits, amylose content and paste properties.**
DMC: dry matter content (%); HCN: cyanogenic compounds content; StC: starch content (%); Aml: amylose content (%); PeV: peak viscosity (cP); MiV: minimum viscosity; BrD: viscosity breakdown (cP); FiV: final viscosity (cP); SeB: setback (cP); TPV: time to peak viscosity (min); PaT: paste temperature (°C). *, **, and *** indicate significant correlations (p < 0.05, p < 0.01, and p < 0.001, respectively).

cassava exhibited HCN content ranging from score 2 to 4.5, while intermediate genotypes ranged between score 4.6 and 6.0 and bitter genotypes ranged from score 6.1 to 9. Moderate to high HCN content genotypes displayed significant positive correlations between dry matter content and starch content. However, in genotypes with low HCN content (sweet), negative correlations were observed between dry matter content and amylose content (-0.23) and between starch content and amylose content (-0.27).

To assess Pearson correlations among the evaluated traits, genotypes were categorized based on starch content (low < 19.60% and high > 19.60%) and dry matter content (low < 34.00% and high > 34.00%). In genotypes with low starch content, significant negative correlations were found between amylose content versus starch content (-0.55) and dry matter content (-0.39). Conversely, genotypes with high starch content exhibited significant but weak correlations (-0.17 to 0.24) with all paste property parameters (Fig 5). Despite variations in the starch content among the clones, the correlations between the parameters of paste properties remained consistent in both direction and magnitude, except for a few specific comparisons. For instance, correlations such as viscosity breakdown × HCN; time to peak viscosity × HCN, amylose content, minimum viscosity, final viscosity, and setback; paste temperature × peak viscosity, setback, and time to peak viscosity.

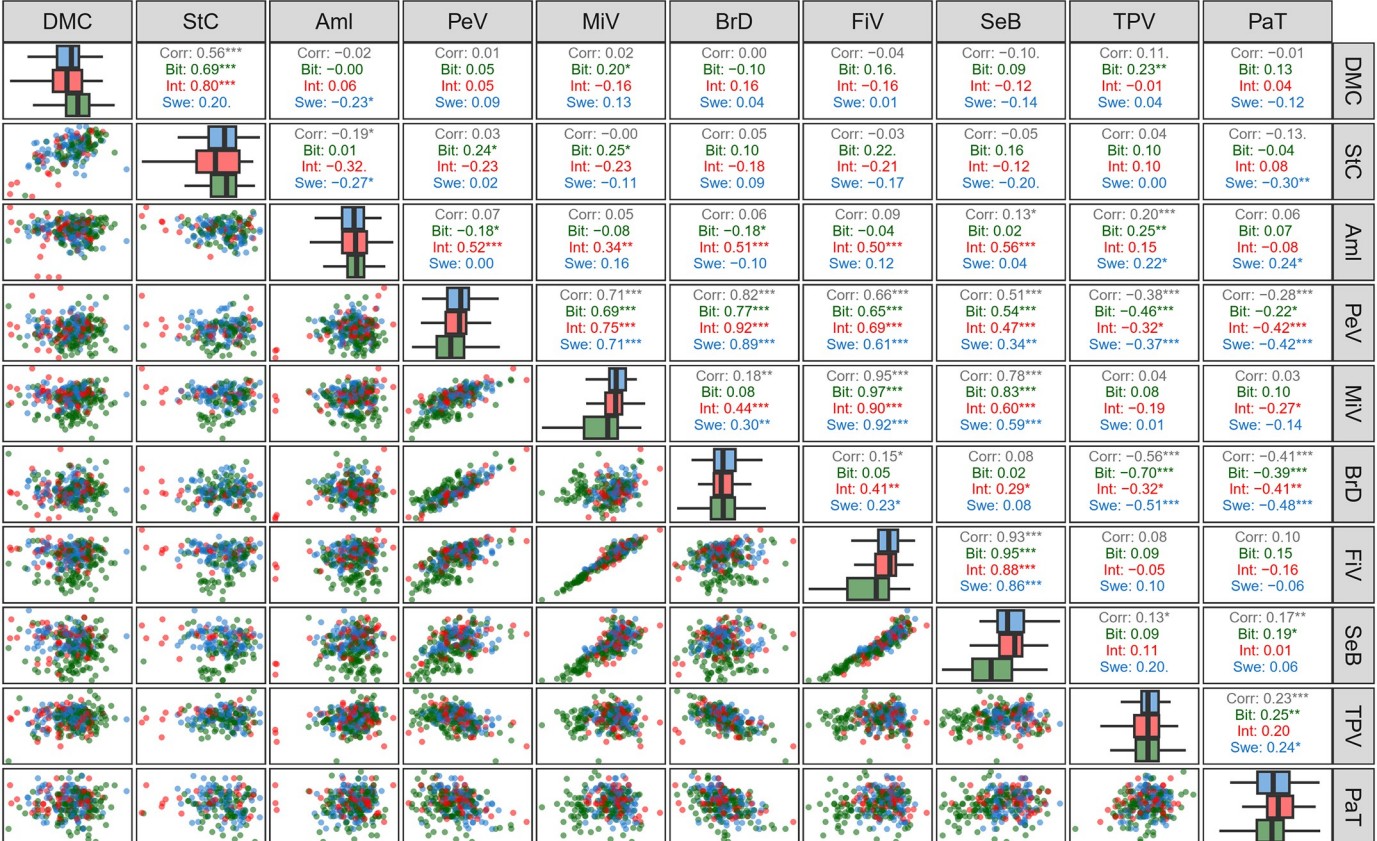

**Fig 4. Pearson correlation of root quality traits, amylose, dry matter content, and starch content in cassava genotypes based on classification of HCN content, that is bitter (Bit), sweet (Swe) and intermediate (Int).** DMC: dry matter content (%); HCN: cyanogenic compounds content; StC: starch content (%); Aml: amylose content (%); PeV: peak viscosity (cP); MiV: minimum viscosity (cP); BrD: viscosity breakdown (cP); FiV: final viscosity (cP); SeB: setback (cP); TPV: time to peak viscosity (min); PaT: paste temperature (°C). *, **, and *** indicate significant correlations (p < 0.05, p < 0.01, and p < 0.001, respectively).

Genotypes with high dry matter content exhibited a significant correlation between dry matter content and HCN (0.22), although of low magnitude. However, in genotypes with low dry matter content, significant correlations were identified between dry matter content × starch content (0.54), of moderate magnitude. In addition, genotypes with low dry matter content were found to correlate with certain paste property traits, including peak viscosity (-0.26), minimum viscosity (-0.23), final viscosity (-0.23), setback (-0.21), and time to peak viscosity (-0.20), albeit with relatively low magnitudes (Fig 6).

Fig 7 presents the network built based on the main significant correlations between paste properties, starch quality, and root quality. Overall, there was a clear separation between paste property and root quality traits. Additionally, strong relationships were identified between paste properties (peak viscosity and viscosity breakdown, final viscosity and setback), as previously shown in the correlation analysis where the correlations between amylose content and paste properties were positive and of high magnitude in breeding populations (ranging from 0.60 to 0.86) compared to the correlations found in the germplasm. On the other hand, the paste temperature trait showed lower association with the other paste property attributes and root quality. Regarding starch quality, a positive association was found with paste properties, such as time to peak viscosity and viscosity breakdown.

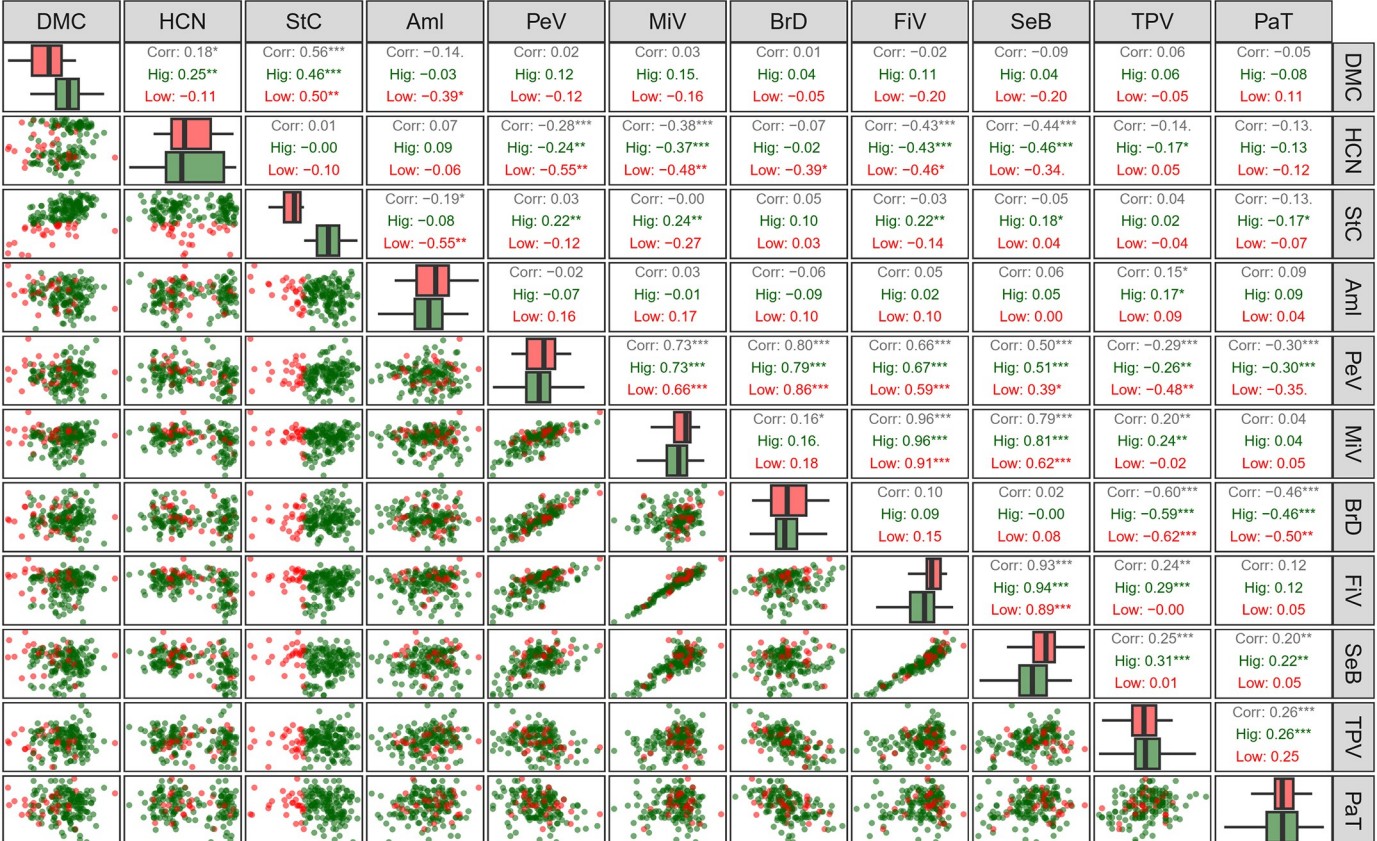

**Fig 5. Pearson phenotypic correlation in cassava genotypes with high (Hig) and low (Low) starch content.** DMC: dry matter content; HCN: cyanogenic compounds content; StC: starch content (%); Aml: amylose content (%); PeV: peak viscosity (cP); MiV: minimum viscosity (cP); BrD: viscosity breakdown (cP); FiV: final viscosity (cP); SeB: setback (cP); TPV: time to peak viscosity (min); PaT: paste temperature (°C). *, **, and *** indicate significant correlations (p < 0.05, p < 0.01, and p < 0.001, respectively).

## 3.3 Discriminant analysis of principal components (DAPC)

DAPC was conducted to identify the genetic relationship among cassava genotypes based on phenotypic data of root quality, amylose content, and pasta properties (Fig 8). Three discriminant functions were retained, explaining 99% of the total genetic variation and allowing the grouping of genotypes into 6 clusters, consisting of 32, 52, 31, 17, 25, and 23 genotypes.

Fig 9 depicts the distributions of phenotypic data concerning root quality for the six groups identified through DAPC analysis. In Group 1, the predominant attributes were median amylose content, low starch, and dry matter content. Group 2 exhibited median levels of amylose, dry matter, and starch content, alongside elevated levels of cyanogenic compounds. Group 3 displayed the widest data variation for both amylose content and cyanogenic compounds. While Group 4 featured median levels of amylose, starch, and dry matter content, it showed the lowest mean and variation in cyanogenic compound content. Group 5 was typified by low amylose content, yet high levels of starch and dry matter. Lastly, Group 6 encompassed genotypes with median amylose and dry matter content, but notably high levels of starch content.

When analyzing the boxplots representing paste properties, a clear differentiation emerges: Genotypes within Group 1 are notably distinguished by their high final viscosity, high retrogradation tendency, and elevated pasting temperatures (Fig 10). In contrast, genotypes in Group 2 tend to exhibit low average values for maximum, minimum, and final viscosity. Meanwhile, Group 3 stands out for its median viscosity values across maximum, minimum,

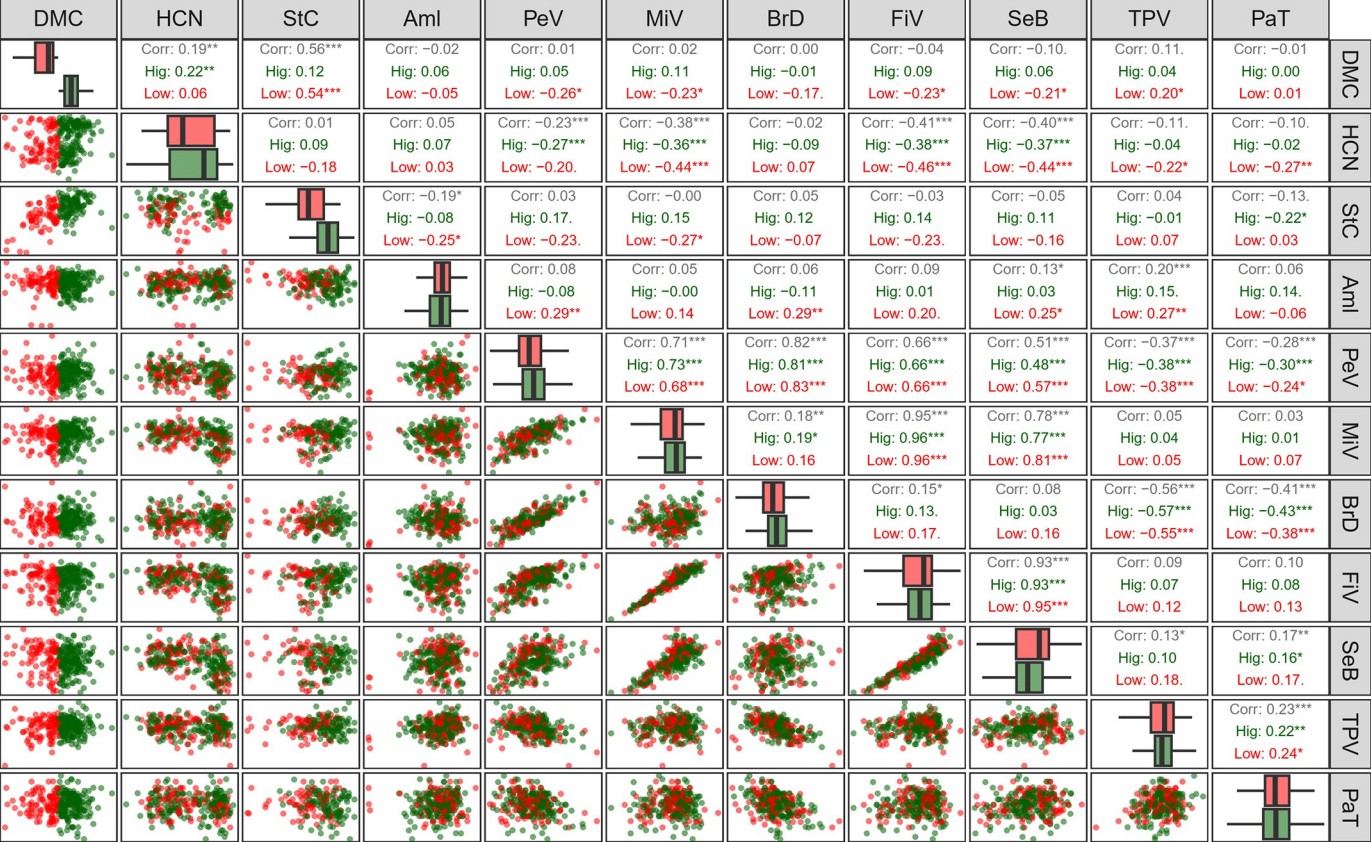

**Fig 6. Pearson correlation in cassava genotypes with high and low dry matter content.** DMC: dry matter content; HCN: cyanogenic compounds content; StC: starch content; Aml: amylose content (%); PeV: peak viscosity (cP); MiV: minimum viscosity (cP); BrD: viscosity breakdown (cP); FiV: final viscosity (cP); SeB: setback (cP); TPV: time to peak viscosity (min); PaT: paste temperature (˚C). *, **, and *** indicate significant correlations (p < 0.05, p < 0.01, and p < 0.001, respectively).

and final measurements, accompanied by a low viscosity breakdown and a notable retrogradation tendency, along with high pasting temperatures.

Genotypes within Group 4 exhibited notably higher values across various parameters (excluding the improved cultivars), including maximum viscosity, minimum viscosity, viscosity breakdown, final viscosity, and lower pasting temperature, compared to other groups. In contrast, genotypes in Group 5 demonstrated, on average, lower values of minimum viscosity, final viscosity, time to reach peak viscosity, and retrogradation tendency (averaging 733.37 cP) in comparison to the other groups. Conversely, Group 6 was distinguished by lower maximum viscosity, viscosity breakdown, and higher pasting temperature compared to the other groups. The majority of genotypes from the breeding population were predominantly clustered within Groups 3 and 4, indicating that these genotypes were grouped due to genetic enhancements, resulting in high average starch content (averaging 20.40%) and dry matter content (averaging 34.32%). The RVA profile of the six groups identified by DAPC is illustrated in Fig 11, providing insight into the average viscoamylogram of each group.

## 3.4 Selection of genotypes based on amylose content

Extremes in amylose content are desirable due to their distinct industrial applications. Therefore, the selection of 15 genotypes with high and low amylose content for recombination in the

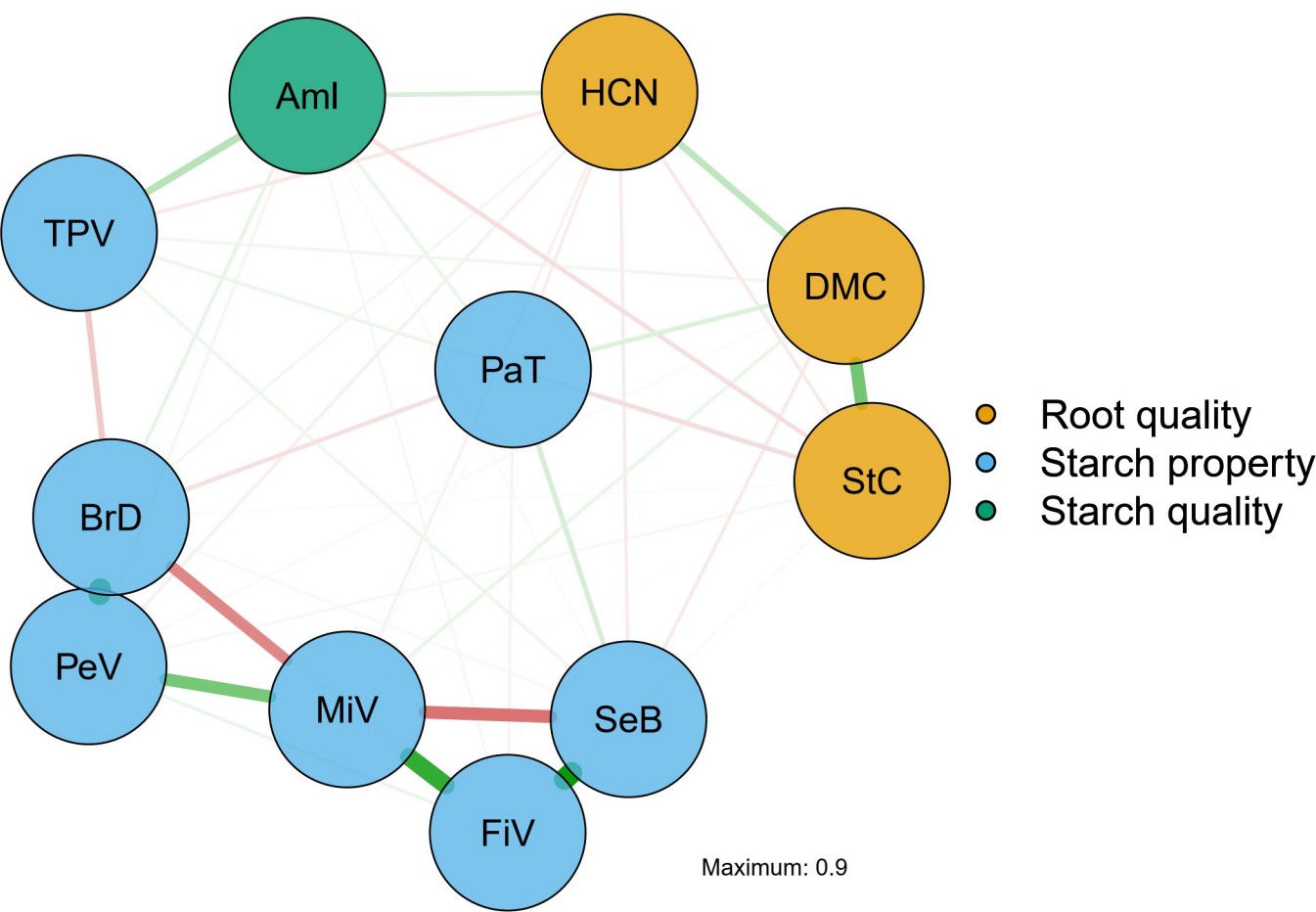

**Fig 7. Regularized network of partial correlations among root quality traits, amylose content, and starch properties evaluated in cassava genotypes.** The thickness of the lines represents the intensity of the correlations, while green and red colors represent positive and negative correlations, respectively. DMC: dry matter content (%); HCN: cyanogenic compounds content; StC: starch content (%); Aml: amylose content (%); PeV: peak viscosity (cP); MiV: minimum viscosity (cP); BrD: viscosity breakdown (cP); FiV: final viscosity (cP); SeB: setback (cP); TPV: time to peak viscosity (min); PaT: paste temperature (˚C).

cassava breeding program was performed. Most of the selected genotypes belong to the species germplasm, except for the clones BR-19F1wx-054-29 and BR-19F2wx-356-2 (Table 3). In the preliminary selection of genotypes with low amylose content (<12%), five genotypes with the highest index selection were ranked. The selection differential of the selected genotypes resulted in an increase in dry matter content (0.19%) and starch (2.06%), and a reduction in HCN by 0.26 compared to 1.53 in genotypes with high amylose content. On the other hand, for genotypes with the highest amylose content (>25%), 10 genotypes were ranked with an average of 27.18% amylose (range from 24.98% to 29.95%). The selection differential of genotypes with high amylose content allowed for an increase in dry matter content (1.23%) and starch (1.54%), as well as a reduction in HCN (score 1.53) (Table 3).

## 4. Discussion

### 4.1 Genetic variability for root quality traits and amylose content in cassava

Estimating phenotypic variation in cassava germplasm is crucial for determining usable genetic variability and assessing potential selection gains. In this study, we observed a wide

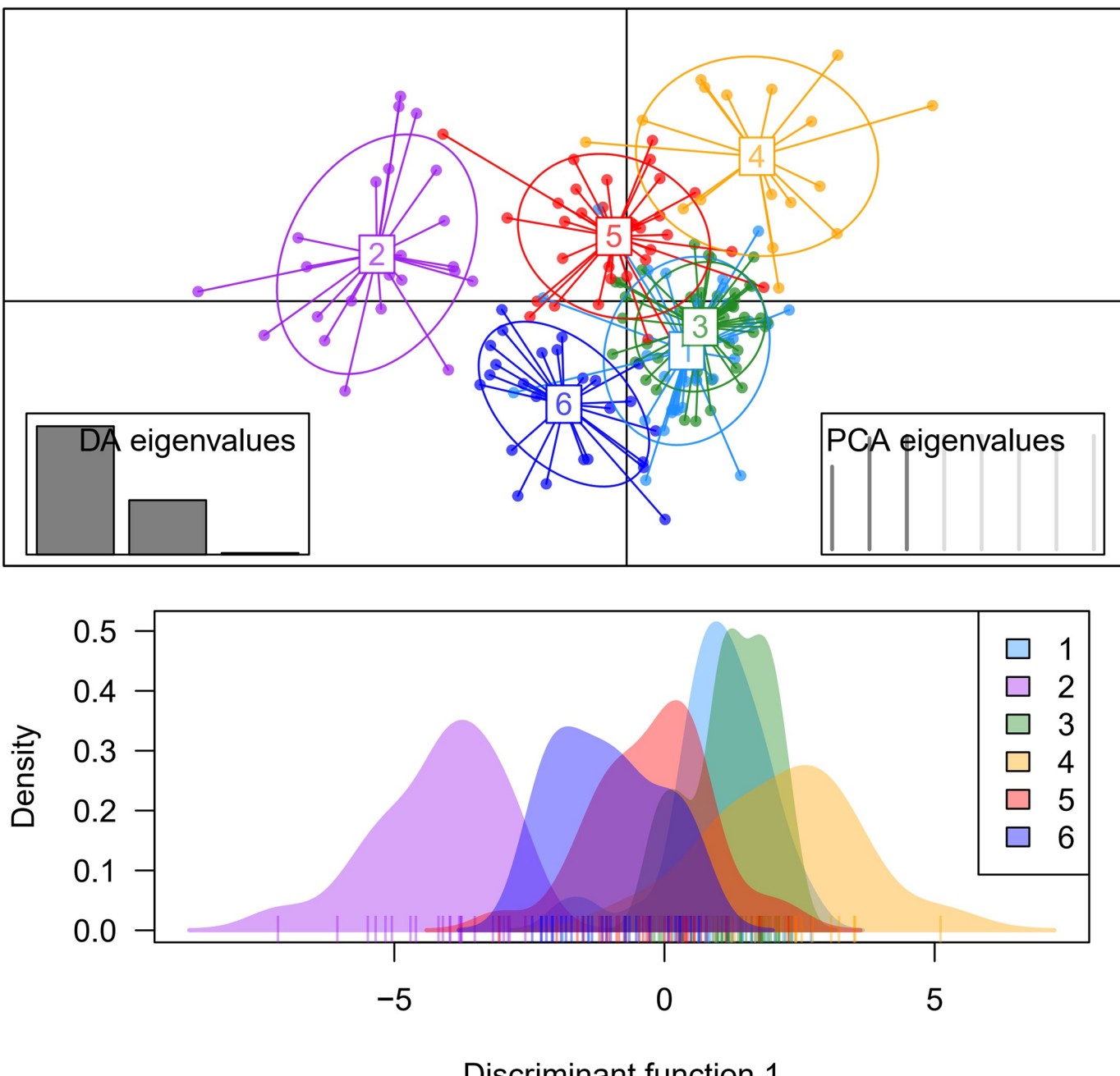

**Fig 8. Discriminant analysis of principal components (DAPC) showing the genetic relationship of cassava genotypes considering phenotypic data on root quality traits, amylose content, and paste properties.**

range of variation in root quality traits, amylose content, and paste properties, which offers opportunities for selecting genotypes suitable for commercial use or as parents in breeding programs.

Concerning root quality traits, the evaluated germplasm exhibited significant variability in dry matter content, ranging from 27.05% to 41.01%, with a mean of 34.75%. These findings align with those reported by Sánchez et al. [10] and Vasconcelos et al. [31], who found

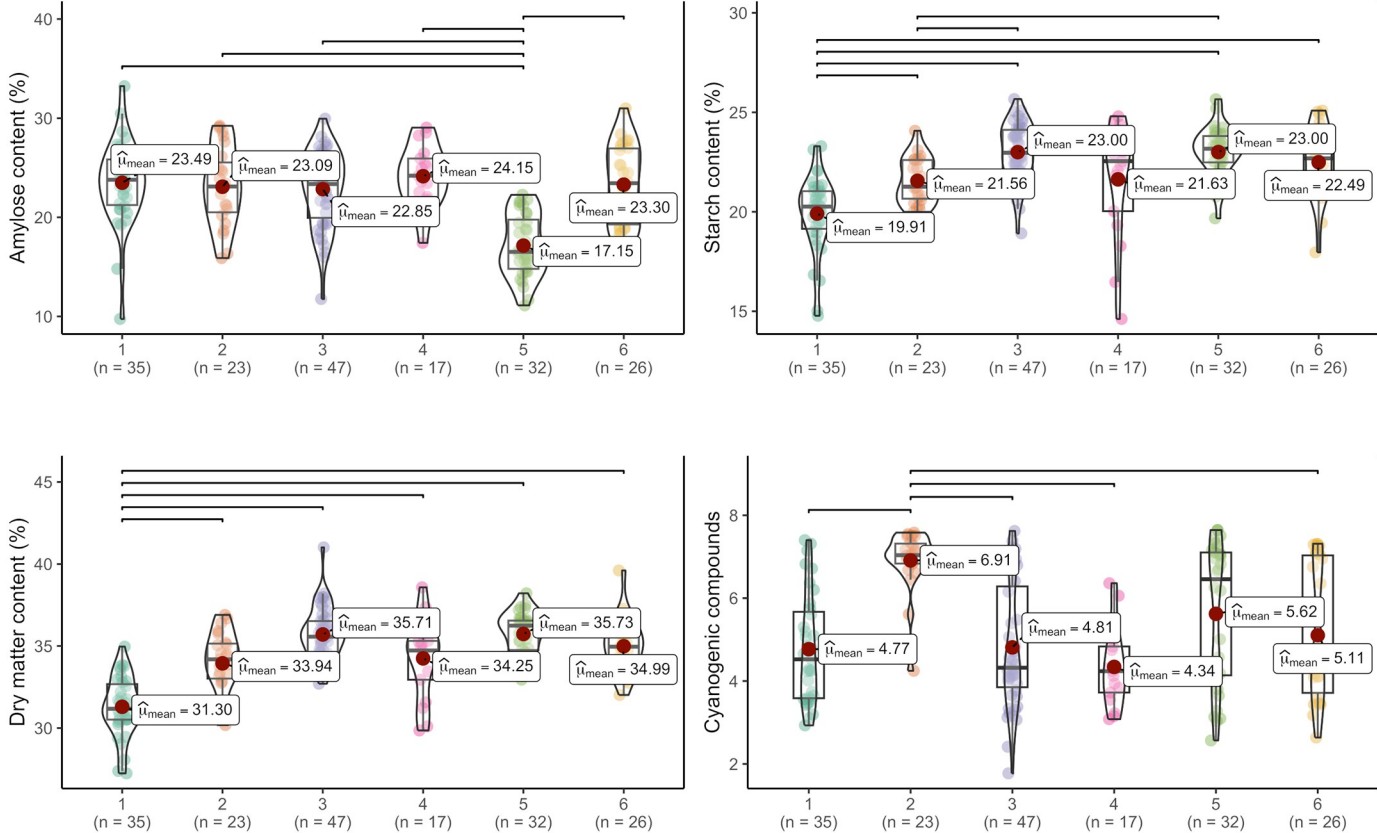

**Fig 9. Boxplot of different groupings of cassava genotypes for root quality-related characteristics.** The bars represent comparisons of significant means (p<0.05).

variations in dry matter content from 14.28% to 48.12% and from 28.01% to 39.70%, respectively, in germplasm accessions of *M. esculenta*. Therefore, while environmental factors strongly influence dry matter content [39], our results indicated a substantial genotypic contribution to this trait's variability.

Similarly, starch content in the roots exhibited considerable variation, ranging from 14.61% to 25.66%, with mean values of 22.18% and 20.49% in the germplasm and breeding population, respectively. These findings are consistent with those of Teixeira et al. [20], who observed starch content variations from 12.96% to 25.92% with a general mean of 19.47% across various cassava genotypes. The strong correlation between dry matter and starch content in cassava roots, as reported by Vasconcelos et al. [31] and Fukuda et al. [30], suggests similar distributions in comparison to this study.

In the context of cyanogenic compounds, the germplasm analyzed encompassed genotypes categorized as sweet (HCN < score 4.50), intermediate (4,5 > HCN < 6.00), and bitter (HCN > score 6.00) according to Fukuda et al. [30]. On average, the population exhibited intermediate levels of HCN, ranging from scores 5.19 to 5.51. Cyanogenic compound content holds particular importance in the development of sweet cassava varieties, predominantly consumed in cooked form, where roots should contain less than 100 mg kg$^{-1}$ of HCN to ensure safety in consumption [40].

The variability in amylose content may influence the quality and texture of starch-based foods. Thus, understanding the correlation between paste properties and amylose content in

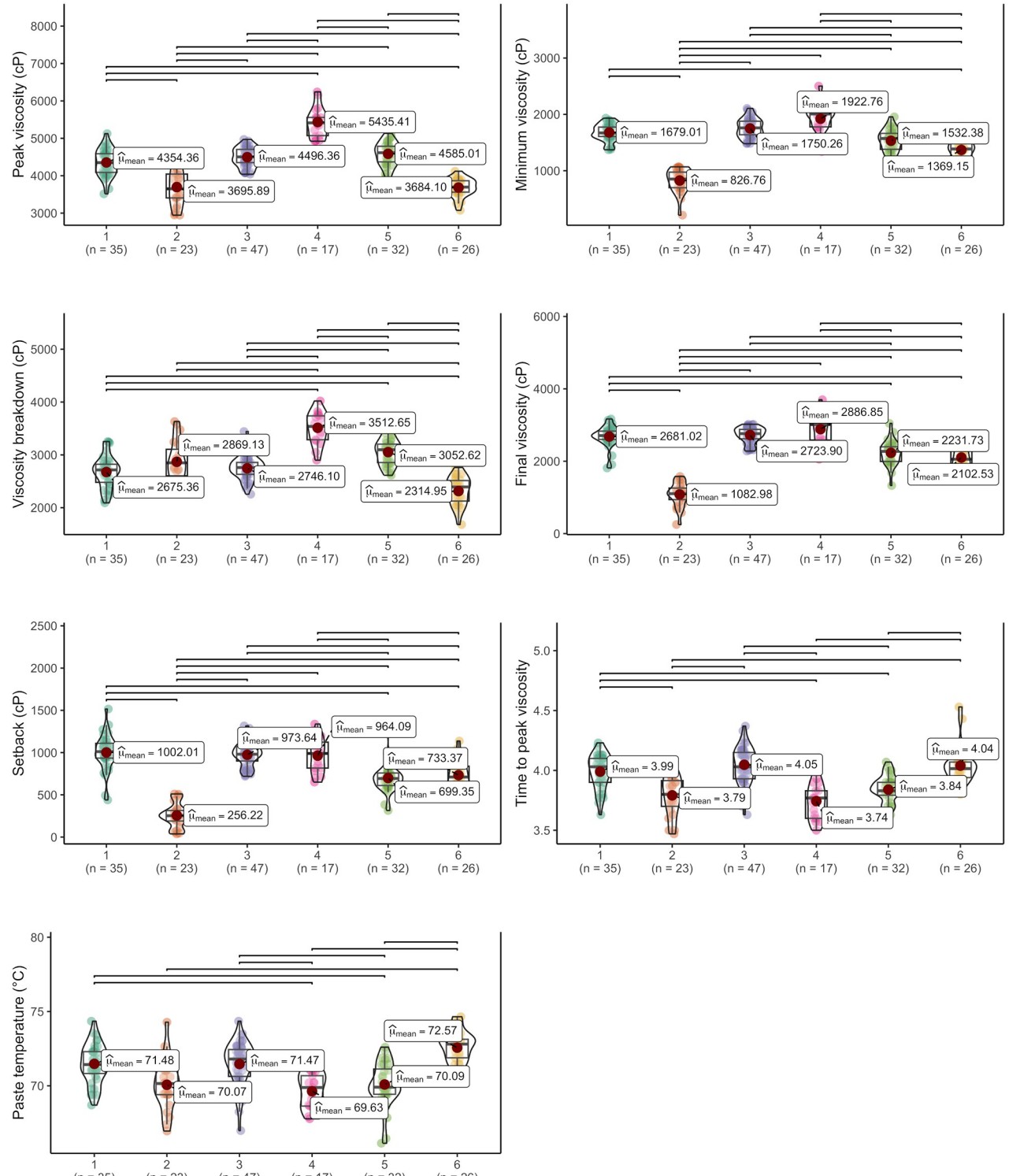

**Fig 10. Boxplot of the diverse groupings of cassava genotypes based on paste properties.** The bars denote comparisons of statistically significant means (p<0.05).

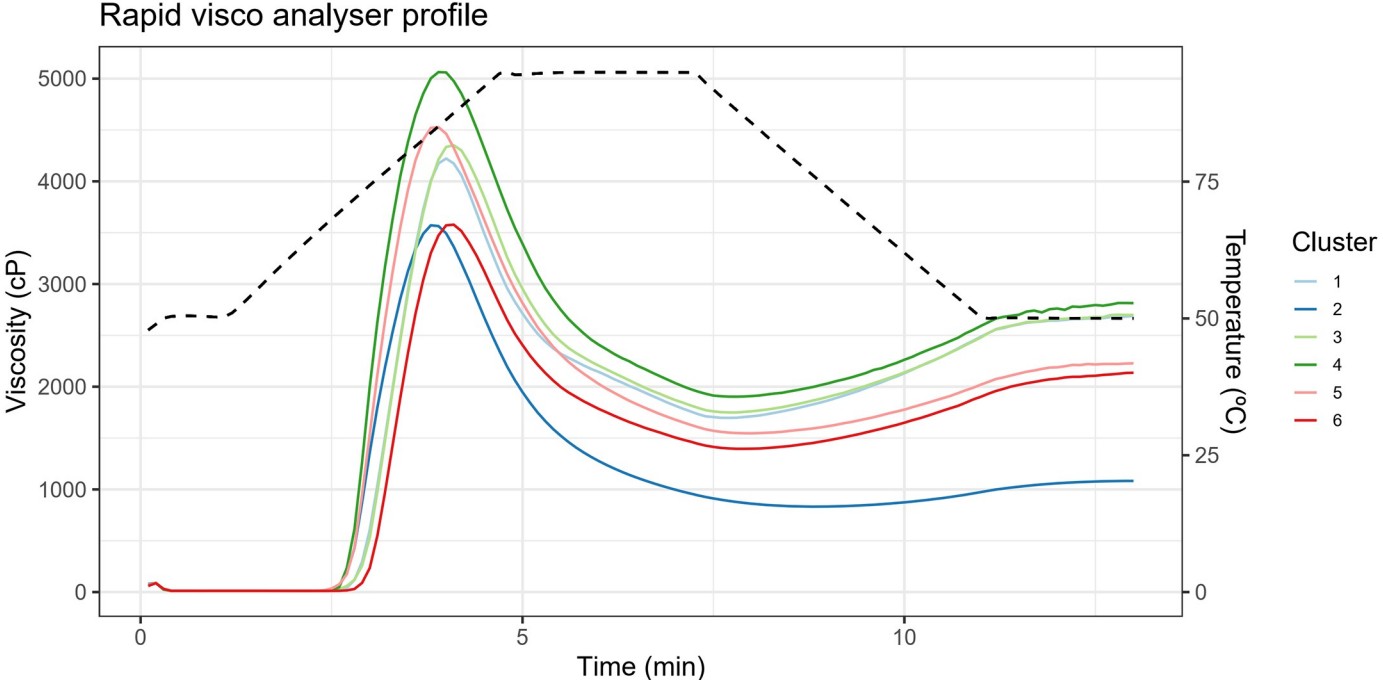

**Fig 11. Rapid visco analyser profile of different clusters obtained by discriminant analysis of principal components analysis of cassava genotypes based on paste properties.**

**Table 3. Lista of selected genotypes for recombination to generate cassava progenies with low and high amylose content.**

| Genotype | Population | DMC | HCN | StC | Rank (Amylose < 12%) |
|---|---|---|---|---|---|
| BGM-0480 | Germplasm | 35.67 | 3.80 | 26.57 | 1 |
| BGM-1042 | Germplasm | 34.26 | 3.66 | 25.28 | 2 |
| BR-19F1wx-054-29 | Improved | 31.84 | 4.97 | 24.16 | 3 |
| BGM-1834 | Germplasm | 30.81 | 4.52 | 23.36 | 4 |
| BR-19F2wx-356-2 | Improved | 32.99 | 5.55 | 18.80 | 5 |
| **Differential of selection** | | 0.19 | -0.26 | 2.06 | |
| Genotype | Population | DMC | HCN | StC | Rank (Amylose > 25%) |
| BGM-2358 | Germplasm | 34.04 | 2.42 | 25.24 | 1 |
| Aipim-Guanambi | Germplasm | 35.36 | 1.77 | 21.12 | 2 |
| BGM-1313 | Germplasm | 35.57 | 3.16 | 22.53 | 3 |
| BGM-1106 | Germplasm | 36.58 | 5.36 | 27.28 | 4 |
| BGM-1148 | Germplasm | 36.84 | 4.83 | 24.80 | 5 |
| BGM-1454 | Germplasm | 36.27 | 4.11 | 22.95 | 6 |
| 2019g-08-53 | Germplasm | 35.52 | 3.43 | 21.59 | 7 |
| BGM-0998 | Germplasm | 35.99 | 4.18 | 22.90 | 8 |
| BGM-0070 | Germplasm | 35.70 | 4.14 | 22.56 | 9 |
| BGM-1044 | Germplasm | 34.77 | 3.31 | 20.71 | 10 |
| **Differential of selection** | | 1.23 | -1.53 | 1.54 | |

DMC: Dry matter content (%); HCN (score 1–9): Cyanogenic compound content; StC: Starch content (%).

cassava is pivotal for guiding genetic improvement programs. Assessing diverse populations exhibiting wide-ranging amylose content is crucial for precisely delineating the relationship between paste properties and for developing more promising industrial starches [6]. Amylose content displays notable variability among various cassava genotypes [8] and serves a critical role in evaluating the quality of products where starch is a primary component [8, 41, 42]. In the current study, amylose values ranged from 0.05 to 33.23%, a broader range than that reported by Teixeira et al. [20], whose variation spanned from 13.29% to 29.91%. Unlike our findings, where amylose content was high among germplasm bank genotypes (22.1% versus 19.8% in the breeding population), Sánchez et al. [10] reported an average amylose content of 20.7%, with no significant discrepancy between local varieties and clones from the breeding program. In their study, amylose content varied from 15.02% to 26.5%. Variations in results can be attributed to several factors, such as genetic differences among cassava populations, environmental conditions, harvest times, laboratory techniques, and post-harvest processing methods like drying and fermentation, which can alter starch structure and pasting behavior.

## 4.2 Genetic variability for paste properties

The viscosity profile of starch is characterized by distinct stages, including paste formation, cooking, and retrogradation [15]. Viscosity is pivotal in characterizing the rheological properties of various products, serving as a measure to evaluate a substance's ability to deform or resist deformation when subjected to tension [43]. Starch paste viscosity is measured over time using RVA, providing insights into gelatinization capacity, paste stability, and other attributes relevant to starch processing [44].

Our study revealed significant phenotypic variation among genotypes in all assessed paste properties. This variability aligns with previous findings on cassava starch [10, 14–19]. Sánchez et al. [10] conducted an extensive investigation into starch paste characteristics, evaluating over 4,000 cassava genotypes from International Center for Tropical Agriculture (CIAT) and identified considerable variability in starch paste properties, particularly maximum viscosity and viscosity breakdown.

In another study, Santos et al. [17] assessed 1031 cassava accessions for phenotypic variation in breeding, focusing on starch paste properties for food applications. They observed pasting temperature variation ranging from 64.1 ˚C to 74.1 ˚C. While Sánchez et al. [10] reported an average pasting temperature of 65.3 ˚C, our study found a slightly higher average (71.1 ˚C) for both the germplasm and breeding population (69.7 ˚C). This variance holds significance, offering potential benefits in the food industry by showcasing a broader range compared to conventional cultivars (63.4 ˚C to 66.9 ˚C) [17], as high pasting temperature in starch indicates increased energy and time requirements for processing, but it can enhance heat stability and texture in certain products. On the contrary, the low pasting temperature indicate low resistance to swelling and easy formation of paste which makes it more suitable for both food and non-food industrial processes because of a decrease in energy costs during production [19].

Starch retrogradation involves the rearrangement during cooling, leading to the formation of organized crystalline structures. This process is highly relevant in starch processing, alongside its thermal properties, directly impacting the texture, stability, and other attributes of products incorporating starch as an ingredient [8]. In the current study, the average retrogradation tendency measured was at 785.8 cP, with a range spanning from 38 to 1516 cP. This value appears notably high when compared to data reported by Sánchez et al. [10] (ranging from -702 to 273 cP) and Onitilo et al. [19] (ranging from 19 to 80 cP). High starch retrogradation implies that the starch will crystallize and become less soluble over time, leading to texture changes such as increased firmness or grittiness in products. This can affect the quality and

shelf-life of foods, potentially making them less desirable for consumers. In contrast, Santos et al. [11] provided specific retrogradation tendency values for each group of evaluated cassava accessions. They examined 881 cassava genotypes from the Brazilian germplasm bank, along with their industrial potential using cassava and waxy maize clones. Notably, two genotypes exhibited a low retrogradation tendency, indicating a high capacity to maintain sensory quality and prolong the shelf life of refrigerated or frozen processed foods. Overall, starches with low retrogradation tendency are preferred as they can uphold the texture and quality of foods for extended durations [45].

## 4.3 Correlations between root quality traits, amylose content, and paste properties

Investigating correlations between root quality traits, amylose content, and paste properties is important for guiding the selective breeding process of new cassava varieties. These correlations enable a deeper understanding of the relationship between root quality traits, especially those dependent on more labor-intensive laboratory analyses such as amylose content and paste properties [6].

In this study, the correlation between root quality traits in a breeding population and a set of germplasm was analyzed. Significant positive correlations with moderate magnitudes were found in two groups of clones: one from the breeding population and the other from the germplasm. Specifically, there was a positive relationship between starch content and dry matter (0.547 for the germplasm population and 0.591 for the breeding population). Other studies have reported a high correlation between starch content and dry matter (0.89) in cassava varieties from southwest Ethiopia [46]. This high correlation is due to the fact that the majority of dried cassava root is composed of carbohydrates, with over 80% being starch [47]. Additionally, there were negative correlations of low magnitude between amylose content and starch content (-0.178 for the germplasm population and -0.222 for the breeding population).

In the breeding population, amylose content showed significant correlations with paste properties, with medium magnitude for peak viscosity (0.610) and minimum viscosity (0.706), and high magnitude for other properties (>0.810), except for pasting temperature where there was no significant correlation. There was a positive relationship between minimum viscosity and peak viscosity (0.129 in the germplasm population), but of low magnitude, while for retrogradation and final viscosity, the correlation of minimum viscosity showed high magnitude (0.795 and 0.957 respectively). In contrast, Mtunguja et al. [48] found no significant differences in amylose content, starch content, or pasting properties among six cassava cultivars. This lack of variation may have been due to the narrow range of amylose content in the landraces studied. On the other hand, Alamu et al. [14] observed significant differences among 400 cassava genotypes and identified positive correlations between dry matter and pasting properties, which is consistent with the findings of the present study.

Although previous research reported no significant correlation between HCN levels and dry matter content [31, 48], we identified that genotypes with intermediate and high levels of HCN exhibited significant and medium-magnitude correlations between dry matter and starch content, with coefficients (0.798 and 0.687, respectively). Negative correlations of medium magnitude were observed between viscosity breakdown and pasting temperature, as well as viscosity breakdown and peak viscosity in genotypes with both high and low starch content. Additionally, there was a low-magnitude positive correlation for peak viscosity and minimum viscosity (0.244), final viscosity (0.286), and retrogradation (0.308). These findings are consistent with Alamu et al. [14], who reported that peak viscosity had a positive and

significant relationship with minimum viscosity (0.60), final viscosity (0.63), and retrogradation (0.40) when analyzing various cassava cultivars in Nigeria.

Amylose content correlated with almost all parameters associated with paste properties, except pasting temperature in the breeding population, indicating a faster pasting process, reflecting in reduced cooking time for genotypes with higher amylose content [20]. In contrast, the germplasm population showed correlations between amylose content only with peak viscosity. Our findings contrast with those of Tappiban et al. [8], who reported a positive correlation (0.69) between the amylose content of cassava cultivars and pasting temperature. However, it is important to note that their study analyzed a limited panel of genotypes (five Thai cultivars), which makes direct comparison with our study challenging.

High correlations were identified between maximum viscosity and viscosity breakdown (0.789 in the germplasm population and 0.986 in the breeding population), possibly due to starch structure and its behavior during processing or cooking, as amylose content directly affects gel formation and stability during cooking [44]. Final viscosity and retrogradation tendency are related to changes in starch characteristics after cooking and cooling; that is, starches with higher amylose content tend to have higher final viscosity and greater retrogradation tendency after cooling, which is relevant for starch-containing food products as it affects texture and consistency after cooling and storage [7, 8]. Meanwhile, pasting temperature demonstrated a weaker association with other pasting property attributes and root quality traits.

## 4.4 Clustering of genotypes based on root quality traits, amylose content, and paste properties

The segregation of cassava genotypes into specific groups can be useful in plant breeding and allocating specific uses in the industry. Correlations between amylose content and paste properties are important as they directly affect food quality and texture [6]. Genotypes belonging to Group 5 exhibit low amylose content and final viscosity, making them ideal for producing products that require stability during freezing, due to their lower tendency to retrogradation [11]. The other groups contain higher amylose levels. Besides that, genotypes in Group 4 exhibit higher final viscosity, likely due to increased hydrogen bonding interactions, and amylose molecules' high tendency for reassociation, which also favors retrogradation [6].

The starch of genotypes in Group 1 shows a high pasting temperature and thus can be used in applications requiring stability and resistance to high temperatures during processing or cooking [44]. Conversely, Groups 1 and 6 show lower retrogradation tendency, making them more suitable for applications where minimizing or avoiding starch crystal formation after cooling is desired [8, 16].

Most genotypes from the breeding population were predominantly grouped in Groups 3 and 4, comprising genotypes that undergone a breeding process resulting in high starch and dry matter content (average of 20.40% and 34.32%, respectively). Additionally, Group 4 exhibited higher values for various parameters such as maximum viscosity, minimum viscosity, and final viscosity. Furthermore, genotypes in Group 4, on average, showed higher viscosity parameters and lower pasting temperature (69.4°C) compared to other groups. Therefore, it can be inferred that Group 4 presents characteristics related to paste properties, such as viscosity and pasting temperature, which may be relevant to produce ready-to-eat foods, such as powdered soup [49], making it an important group to advance the breeding program.

## 5. Conclusion

The phenotypic variability and correlations between amylose content and starch paste properties of different cassava genotypes were analyzed. The results demonstrated that improved

genotypes showed more efficient and consistent outcomes regarding paste properties, providing promising potential for the production of cassava-derived foods and products with desired characteristics. This advancement in understanding the relationship between amylose content, starch content, dry matter, and paste properties can significantly contribute to the agricultural and agro-industrial sector, enabling more precise selection of cassava genotypes to meet the demands of the food and processing industry.

## Supporting information

**S1 Table. Origin of cassava genotypes with their classification regarding cyanogenic compound content (HCN), dry matter content (DMC), and starch content (StC).**
(DOCX)

## Author Contributions

**Conceptualization:** Luciana Alves de Oliveira, Eder Jorge de Oliveira.

**Data curation:** Natalia Rocha Ribeiro, Massaine Bandeira e Sousa, Luciana Alves de Oliveira.

**Formal analysis:** Natalia Rocha Ribeiro, Massaine Bandeira e Sousa.

**Funding acquisition:** Eder Jorge de Oliveira.

**Investigation:** Natalia Rocha Ribeiro.

**Methodology:** Natalia Rocha Ribeiro, Luciana Alves de Oliveira.

**Project administration:** Eder Jorge de Oliveira.

**Resources:** Eder Jorge de Oliveira.

**Supervision:** Massaine Bandeira e Sousa, Luciana Alves de Oliveira, Eder Jorge de Oliveira.

**Writing – original draft:** Natalia Rocha Ribeiro.

**Writing – review & editing:** Massaine Bandeira e Sousa, Luciana Alves de Oliveira, Eder Jorge de Oliveira.

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
