## [Decision Letter · Decision Letter 0]

16 Jul 2024

PONE-D-24-09179Variability of amylose content and its correlation with the paste properties of cassava starchPLOS ONE

Dear Dr. de Oliveira,

Thank you for submitting your manuscript to PLOS ONE. After careful consideration, we feel that it has merit but does not fully meet PLOS ONE’s publication criteria as it currently stands. Therefore, we invite you to submit a revised version of the manuscript that addresses the points raised during the review process.

 Please, check the few concerns submitted by the reviewers, which may enhance manuscript's readability. Please submit your revised manuscript by Aug 30 2024 11:59PM. If you will need more time than this to complete your revisions, please reply to this message or contact the journal office at plosone@plos.org. Please include the following items when submitting your revised manuscript:A rebuttal letter that responds to each point raised by the academic editor and reviewer(s). You should upload this letter as a separate file labeled 'Response to Reviewers'.A marked-up copy of your manuscript that highlights changes made to the original version. You should upload this as a separate file labeled 'Revised Manuscript with Track Changes'.An unmarked version of your revised paper without tracked changes. You should upload this as a separate file labeled 'Manuscript'.If applicable, we recommend that you deposit your laboratory protocols in protocols.io to enhance the reproducibility of your results. Protocols.io assigns your protocol its own identifier (DOI) so that it can be cited independently in the future. For instructions see: https://journals.plos.org/plosone/s/submission-guidelines#loc-laboratory-protocols. Additionally, PLOS ONE offers an option for publishing peer-reviewed Lab Protocol articles, which describe protocols hosted on protocols.io. Read more information on sharing protocols at https://plos.org/protocols?utm_medium=editorial-email&utm_source=authorletters&utm_campaign=protocols.

We look forward to receiving your revised manuscript.

Kind regards,

Santiago Casado Rojo, Ph.D.

Academic Editor

PLOS ONE

Journal Requirements:

2. In your Methods section, please provide additional information regarding the permits you obtained for the work. Please ensure you have included the full name of the authority that approved the field site access and, if no permits were required, a brief statement explaining why

4. Please ensure that you refer to Figure 6 in your text as, if accepted, production will need this reference to link the reader to the figure.

Reviewers' comments:

Reviewer's Responses to Questions

**Comments to the Author**

1. Is the manuscript technically sound, and do the data support the conclusions?

Reviewer #1: Yes

Reviewer #2: Yes

2. Has the statistical analysis been performed appropriately and rigorously? 

Reviewer #1: Yes

Reviewer #2: Yes

3. Have the authors made all data underlying the findings in their manuscript fully available?

Reviewer #1: Yes

Reviewer #2: Yes

4. Is the manuscript presented in an intelligible fashion and written in standard English?

Reviewer #1: Yes

Reviewer #2: Yes

5. Review Comments to the Author

Reviewer #1: The objective of the article Variability of amylose content and its correlation with the paste properties of cassava starch was to examine the genetic diversity related to amylose content in the cassava germplasm bank and explore its correlation with the features of starch paste in a population that exhibits significant differences in root yield and characteristics. I find the article suitable for publishing after minor revision.

Reviewer #2: The manuscript is technically sound and the data supports the conclusions. The methodology section was well described, but needs some updates. The statistical analysis was well performed but the p value needs to be provided. All data used in the study have been provided by the authors. The manuscript is well written but some parts need to be re-written for clarification as indicated in my comments.

6. PLOS authors have the option to publish the peer review history of their article (what does this mean?). If published, this will include your full peer review and any attached files.

Reviewer #1: **Yes: **Valentina Nikolić

Reviewer #2: **Yes: **Dr. Fidelis C.K. Ocloo

---

## [Author Response · Author response to Decision Letter 0]

6 Aug 2024

PONE-D-24-09179

Variability of amylose content and its correlation with the paste properties of cassava starch

Reviewer 1:

I find the article suitable for publishing after minor revision.

These are my comments:

Page 4, line 89: Correlations between pasting properties and amylose content were found on wheat, as well (see https://doi.org/10.1111/jfpp.15805).

Response: We have added the requested citation

Page 6, line 132: Please translate the words in the formula into English.

Response: Ok, thank you for letting us know about the need for the translation.

Reviewer 2:

All the minor changes requested in the PDF file have been made in the new document (in red).

---

## [Decision Letter · Decision Letter 1]

15 Aug 2024

Variability of amylose content and its correlation with the paste properties of cassava starch

PONE-D-24-09179R1

Dear Dr. de Oliveira,

We’re pleased to inform you that your manuscript has been judged scientifically suitable for publication and will be formally accepted for publication once it meets all outstanding technical requirements.

Kind regards,

Santiago Casado Rojo, Ph.D.

Academic Editor

PLOS ONE

Additional Editor Comments (optional):

Reviewers' comments:

Reviewer's Responses to Questions

**Comments to the Author**

1. If the authors have adequately addressed your comments raised in a previous round of review and you feel that this manuscript is now acceptable for publication, you may indicate that here to bypass the “Comments to the Author” section, enter your conflict of interest statement in the “Confidential to Editor” section, and submit your "Accept" recommendation.

Reviewer #1: All comments have been addressed

Reviewer #2: All comments have been addressed

2. Is the manuscript technically sound, and do the data support the conclusions?

Reviewer #1: Yes

Reviewer #2: Yes

3. Has the statistical analysis been performed appropriately and rigorously? 

Reviewer #1: Yes

Reviewer #2: Yes

4. Have the authors made all data underlying the findings in their manuscript fully available?

Reviewer #1: Yes

Reviewer #2: Yes

5. Is the manuscript presented in an intelligible fashion and written in standard English?

Reviewer #1: Yes

Reviewer #2: Yes

6. Review Comments to the Author

Reviewer #1: The authors have corrected all of the suggested issues properly. The manuscript is now suitable for publishing in its present form.

Reviewer #2: I am satisfied with the responses to my comments. Your manuscript is highly commendable. I wish to congratulate the authors for the meticulous work done.

7. PLOS authors have the option to publish the peer review history of their article (what does this mean?). If published, this will include your full peer review and any attached files.

Reviewer #1: **Yes: **Valentina V. Nikolić

Reviewer #2: **Yes: **Dr. Fidelis C.K. Ocloo

---

## [Editor Report · Acceptance letter]

21 Aug 2024

PONE-D-24-09179R1 

PLOS ONE

Dear Dr. de Oliveira, 

I'm pleased to inform you that your manuscript has been deemed suitable for publication in PLOS ONE. Congratulations! Your manuscript is now being handed over to our production team.

Kind regards, 

on behalf of

Dr. Santiago Casado Rojo 

Academic Editor

PLOS ONE